# Molecular Characterization and Phylogenetic Analysis of *Outer membrane protein P2* (*OmpP2*) of *Glaesserella* (*Haemophilus*) *parasuis* Isolates in Central State of Peninsular Malaysia

**DOI:** 10.3390/pathogens12020308

**Published:** 2023-02-12

**Authors:** Chee Yien Lee, Hui Xin Ong, Chew Yee Tan, Suet Ee Low, Lai Yee Phang, Jyhmirn Lai, Peck Toung Ooi, Michelle Wai Cheng Fong

**Affiliations:** 1Department of Veterinary Clinical Studies, Faculty of Veterinary Medicine, Universiti Putra Malaysia, UPM, Serdang 43400, Malaysia; 2Department of Bioprocess Technology, Faculty of Biotechnology and Biomolecular Sciences, Universiti Putra Malaysia, UPM, Serdang 43400, Malaysia; 3Department of Veterinary Medicine, College of Veterinary Medicine, National Chiayi University, Chiayi City 60004, Taiwan

**Keywords:** *Glaesserella (Haemophilus) parasuis*, Malaysia, outer membrane protein (*OmpP2*), phylogenetic tree

## Abstract

*Glaesserella (Haemophilus) parasuis*, the etiological agent of Glässer’s disease, is an economically significant pathogen commonly associated with serofibrinous polyserositis, arthritis, fibrinous bronchopneumonia and/or meningitis. This study is the first attempt to molecularly characterize and provide a detailed overview of the genetic variants of *G. parasuis* present in Malaysia, in reference to its serotype, virulence-associated trimeric autotransporters (*vtaA*) gene and outer membrane protein P2 (*OmpP2*) gene. The *G. parasuis* isolates (*n* = 11) from clinically sick field samples collected from two major pig producing states (Selangor and Perak) were selected for analysis. Upon multiplex PCR, the majority of the isolates (eight out of 11) were identified to be serotype 5 or 12, and interestingly, serotypes 3, 8 and 15 were also detected, which had never been reported in Malaysia prior to this. Generally, virulent *vtaA* was detected for all isolates, except for one, which displayed a nonvirulent *vtaA*. A phylogenetic analysis of the *OmpP2* gene revealed that the majority of Malaysian isolates were clustered into genotype 1, which could be further divided into Ia and Ib, while only one isolate was clustered into genotype 2.

## 1. Introduction

Glässer’s disease is caused by the previously known *Haemophilus parasuis*. It has recently been reclassified to *Glaesserella parasuis* after a series of detailed phylogenomic analysis, in honor of K. Glässer, who first described the disease in pigs, caused by a fastidious, Gram-negative coccobacillus isolated from the respiratory tract or systemic sites of swine [1]. *G. parasuis* may be present in most commercial pig farms as a colonizer of the upper respiratory tract of healthy pigs. It can be detected in the nasal cavity as early as 2 days after birth in nonvaccinated herds, as part of their normal respiratory microbiota [2]. On the other hand, opportunistic *G. parasuis* strains can lead to the onset of lower respiratory tract disease (involved in porcine respiratory disease complex) after weaning, or other clinical presentations characterized by serofibrinous polyserositis, arthritis, and/or meningitis. This often sequels with high morbidity and mortality, especially in the naive swine population, causing significant economic impact to the pig production industry [3].

To outline effective surveillance, control and preventive measures, various efforts in the characterization and identification of *G. parasuis* virulence indicators have been described over the years, gaining a broader understanding of the disease’s nature. Typically, *G. parasuis* is characterized by its serotype (1 to 15) [4,5]. The correlation between serotype and virulence is doubtful; there are conflicting reports on different strains of the same serotype showing variable degrees of symptoms, or different pigs challenged by the same strain showing variable symptoms [6,7,8,9]. Having said that, molecular typing or sequencing methods present a major advancement for epidemiological and vaccine studies, where virulence genes, or even single gene point mutations, can be identified [6,10]. For example, molecular pathotype (i.e., the leader sequence (LS)-PCR classification of virulence and nonvirulence based on *vta*-locus) [11,12], and also by other molecular genotyping methods [6,8,9]. Furthermore, the molecular pathotyping method can be performed easily and correspond well to clinical background information [12].

The virulence-associated trimeric autotransporter (*vtaA*) is one important virulent gene involved in adherence to the host cell and extracellular protein during an infection [13,14]. The *vtaA* genes of *G. parasuis* are differentiated into three groups based on the *YadA* domains; *vtaA*1 and *vtaA*2 are associated with virulence and are a potential indicator of virulent isolates, while *vtaA*3 is highly conserved and is present in all strains [15]. A leader sequence (LS)-PCR based on the extended signal peptide region (ESPR) of the *vtaA*, reported by Galofré-Milà et al. (2017), distinguishes nonvirulent *G. parasuis* commensals from the potentially disease-causing virulent isolates, enabling the prediction of virulence potential of *G. parasuis* [11].

Aside from the trimeric autotransporters (*vtaA*) antigen, the outer membrane protein P2 (*OmpP2*) is amongst the protective antigen identified for *G. parasuis* subunit vaccine developments [16,17,18]. The *OmpP2* gene is the most abundant protein amongst the outer membrane protein, highly conserved and abundant in Gram-negative bacteria [19,20]. It could be divided into two genotypes, which are genetic type-I and type-II [21]. The length of the *OmpP2* gene of genetic type-I ranges from 1077 bp to 1095 bp, while the length of the *OmpP2* gene of genetic type-II ranges from 1167 bp to 1203 bp [22]. The *OmpP2* gene is believed to be associated with the virulence of *G. parasuis,* as it is able to induce proinflammatory cytokine mRNA expression in porcine alveolar macrophages (PAM) [23,24]. The *OmpP2* is considered an immunodominant porin, where it can be targeted as a potential vaccine candidate, subjected to immunogenic characterization, pathogenesis studies, and also serve as a diagnostic antigen [25,26,27,28].

In this study, to characterize the *G. parasuis* isolates of diseased pigs from major pig producing states (Selangor and Perak) in Malaysia, we applied molecular PCR-based and sequencing methods, which were: molecular serotyping using serotype-specific primers, as described by Howell et al. (2015); molecular pathotyping based on the leader sequence (LS)-PCR of *vtaA* gene; as well as molecular genotyping via the phylogenetic analysis of the outer membrane protein P2 (*OmpP2*) gene.

## 2. Materials and Methods

### 2.1. Bacteria Isolates

The central states of Perak and Selangor contributes the highest pig production in Peninsular Malaysia. Farm sizes from both states have sow numbers from 100 to 4000. Due to the pandemic and the threat of African Swine Fever, access to farms, or farms submitting samples specifically for bacteria isolation, was limited. There were 28 *Glaesserella parasuis* isolates retrievable from archive samples (2018–2020; *n* = 24) and recent clinical cases (2021–2022; *n* = 4) presented to the swine unit of the Faculty of Veterinary Medicine, Universiti Putra, Malaysia. The isolation of *G. parasuis* from clinical samples was only performed upon special request by farmers and was not a common routine procedure. The isolation of *G. parasuis* was carried out by inoculating samples submitted on plated chocolate agar (Oxoid, USA), identified by conventional PCR, and positive isolates were preserved at 10% skim milk kept in −80 °C.

The skim milk preserved isolates were subcultured by inoculating 5 µL of skim milk containing *G. parasuis* on plated chocolate agar (Thermo Fisher Scientific, Inc., Waltham, MA, USA) and incubated in a candle jar (which provided a condition of approximate 5% CO_2_) at 37 °C for 24 to 48 h. A loopful of bacteria colonies from the pure culture was suspended into 100 µL deionized water reaching the turbidity of 0.5 McFarland standard. DNA was extracted from the bacteria suspension using a column-based extraction kit (DNeasy^®^ Blood and Tissue Kit, Qiagen, Hilden, North Rhine-Westphalia, Germany) performed according to the manufacturer’s protocol specified for Gram-negative bacteria. The extracted DNA was used for the subsequent PCR assays. To confirm that the bacteria colonies were indeed *G. parasuis*, a conventional PCR assay was carried out using a published species-specific primer of *G. parasuis* [4]. Briefly, a 20 µL PCR reaction mixture of 12.5 µL 2× Mytaq™ Red Mix PCR master mix (Meridian Bioscience^®^, Cincinnati, Ohio, USA), 0.5 nM of forward and reverse primer each, 2 µL (50–100 ng/µL) template and 3.5 µL nuclease-free water was prepared for each sample. The PCR assay was performed at 95 °C for 1 min, followed by 30 cycles of 95 °C for 15 s, 57 °C for 30 s and 72 °C for 30 s, and a final extension of 72 °C for 5 min. All PCR products were subjected to 2% agarose gel electrophoresis in TAE buffer 80 V for 45 min, using a 100-bp molecular weight marker (Qiagen, Hilden, North Rhine-Westphalia, Germany) as a guide. Electrophoresed gels were visualized using a UV transilluminator and gel documentation system (Syngene, Frederick, MD, USA). A PCR band at 275 bp denoted sample was positive of *G. parasuis*.

### 2.2. Multiplex PCR Assay for Serotyping

The isolates were further serotyped by the molecular method published by Howell et al., 2015, following modifications by Schuwerk et al., 2020 [12], which divided the assay into two sets of multiplex PCR (mPCR). The PCR reactions were set up consisting of 12.5 µL 2× Mytaq™ HS Red Mix PCR master mix (Meridian Bioscience^®^, Cincinnati, OH, USA), 0.5 nM of forward and reverse primer each, 2 µL (50–100 ng/µL) template and nuclease-free water topped-up to a volume of 25 µL. The PCR assay was performed at 95 °C for 1 min, followed by 30 cycles of 95 °C for 15 s, 58 °C for 30 s and 72 °C for 30 s, and a final extension of 72 °C for 5 min. All PCR products were subjected to 2% agarose gel electrophoresis in TAE buffer 80 V for 45 min, using a 100-bp molecular weight marker (Qiagen, Hilden, North Rhine-Westphalia, Germany) as a guide. Electrophoresed gels were visualized using a UV transilluminator and gel documentation system (Syngene, Frederick, MD, USA). The PCR bands were measured using the Image Lab software v6.1 (Bio-rad Laboratories, Inc., Hercules, CA, USA).

### 2.3. Conventional PCR Assay for vtaA Gene Identification

One forward and two reverse primers, outlined by Galofré-Milà et al., 2017 [11], were used to distinguish the *vtaA* gene of *G. parasuis* into virulent and nonvirulent strains. The PCR reactions were set up comprising of 12.5 µL 2× Mytaq™ HS Red Mix PCR master mix (Meridian Bioscience^®^, Cincinnati, OH, USA), 0.5 nM of forward and reverse primer each, 2 µL (50–100 ng/µL) template and nuclease-free water topped-up to a volume of 25 µL. The PCR assay was performed at 95 °C for 1 min, followed by 30 cycles of 95 °C for 15 s, 54 °C for 30 s, 72 °C for 30 s and a final extension of 72 °C for 5 min. The PCR products were subjected to 2% agarose gel electrophoresis in TAE buffer 80 V for 45 min, using a 100-bp molecular weight marker (Qiagen, Hilden, North Rhine-Westphalia, Germany) as a guide. Electrophoresed gels were visualized using a UV transilluminator and gel documentation system (Syngene, Frederick, MD, USA). The PCR bands were measured using the Image Lab software v6.1 (Bio-rad Laboratories, Inc., Hercules, CA, USA).

### 2.4. The PCR Amplification and Bioinformatic Analysis of OmpP2 Gene

The extracted DNA were subjected to conventional PCR assay targeting the *G. parasuis OmpP2* gene using primers described by Li et al., 2012 [27]. Briefly, the PCR reactions were set up consisting of 12.5 µL 2× Mytaq™ Red Mix PCR master mix (Meridian Bioscience^®^, Cincinnati, OH, USA), 0.5 nM of primer each, 2 µL template and nuclease-free water topped-up to a volume of 25 µL. The PCR assay was performed at 95 °C for 1 min, followed by 30 cycles of 95 °C for 15 s, 54 °C for 30 s, an extension at 72 °C for 30 s and a final extension of 72 °C for 5 min. The PCR products were subjected to 2% agarose gel electrophoresis in TAE buffer 80 V for 45 min, using a 100-bp molecular weight marker (Qiagen, Hilden, North Rhine-Westphalia, Germany) as a guide. Electrophoresed gels were visualized using a UV transilluminator and gel documentation system (Syngene, Hercules, CA, USA).

The PCR product displaying a single band at approximate 1100 bp was purified using MEGAquick-spinTM Plus total fragment DNA purification kit (iNtRON Biotechnology, Seongnam-si, Gyeonggi-do, South Korea) and outsourced to a sequencing company (Macrogen Asia Pacific Pte. Ltd., Singapore). Samples were sequenced for both forward and reverse strands via Sanger sequencing. Only good quality sequences with quality scores ≥40 were analyzed. Consensus sequences were obtained by aligning the forward sequence and the reverse complement of the reverse sequences using the Mega 11 software [29]. The sequences were analyzed using bioinformatic tools, such as the Mega 11 software for phylogenetic tree construction, and the Clustal Omega (EMBL-EBI) web-based tool for pairwise similarity comparison [29,30]. The sequences were deposited into the NCBI GenBank; their accession numbers are listed in Table 1.

## 3. Results

### 3.1. Bacteria Isolates

Among the 28 retrievable samples, 11 *G. parasuis* isolates were selected from seven larger producing farms with sow number ranges from 300 to 4000. One to three isolates were selected from eight pigs with different isolation sites, namely, lung, pleural swab, brain, peritoneal swab, joint and tonsil.

### 3.2. The Molecular Serotyping of G. Parasuis

The *G. parasuis* isolates were serotyped via mPCR with serotype-specific primers, as shown in Figure 1a,b. It is noteworthy that the mPCR used can distinguish between 14 out of 15 previously described serovars, except serovar 5 and serovar 12, which were detected by the same pair of primers [4].

Serotype 5 or 12 (73%) was found to be the most prevalent serotype, followed by serotypes 3 (9%), 8 (9%) and 15 (9%). These were isolated from cases with polyserositis (73%), arthritis (45%) and fibrinous pneumoniae (18%). There were also multiple *G. parasuis* serotypes isolated from one infected pig showing clinical signs of polyserositis from Selangor, namely, serotype 5 or 12 (GP/UPM/MY005) and serotype 15 (GP/UPM/MY006). Interestingly, serotype 15 was isolated from the tonsils, whereas serotype 5 or 12 was isolated from brain samples in that single animal.

### 3.3. Virulence-Associated Trimeric Autotransporters (vtaA) Gene

The gel electrophoresis of PCR assay revealed that only one isolate (GP/UPM/MY006) displayed a nonvirulent *vtaA* band (208 bp) as shown in Figure 2a and 10 out of 11 isolates displayed virulent *vtaA* bands (179 bp) as shown in Figure 2b. The isolates that displayed virulent *vtaA* were all sampled from clinical sites, e.g., lung, brain and joint, while the only nonvirulent *vtaA* isolate was isolated from a carrier site, which was the tonsil of a clinically ill pig.

### 3.4. Genotyping via OmpP2 Gene

#### 3.4.1. Amplification of *OmpP2* Gene

The primers used in this study worked well at 54 °C temperature to amplify the *OmpP2* gene and produced a target band estimated of 1100 bp (Figure 3) for all 11 isolates, based on PCR optimization using gradient annealing temperature. The 1100 bp region of the *OmpP2* gene amplified by the primers, as evidenced by the NCBI BLAST search results of the sequenced PCR product.

#### 3.4.2. The Bioinformatic Analysis of *G. parasuis OmpP2* Gene Sequence

The sequences of the *OmpP2* gene of *G. parasuis* isolates of Malaysia vary in length from 1077 to 1191 nucleotides. Pairwise comparison using Clustal Omega revealed Malaysia isolates shared a 94.25% to 100% homologous identity among each other as listed in Appendix A. In reference to the other 81 *G. parasuis* strains, Malaysian isolates shared a high homology of 92.98% to 99.87%, except for two strains, Hs-DY06 and Hs-DY13 of China (59.65–65.19%), which is unexpectedly different to the other 79 reference strains. In comparison to *Actinobacillus pleuropneumonia* and *Haemophilus influenzae*, the *OmpP2* gene was indeed distinct to *G. parasuis* with only 43.83–55.94% homology.

In exclusion to the above-mentioned two distinct *G. parasuis* reference strains (Hs-DY06 and Hs-DY13), a total of 79 reference sequences of the *G. parasuis*
*OmpP2* gene were selected for the construction of a phylogenetic tree (Figure 4). The sequences were divided into two clusters, which corresponded with the grouping of genetic type-I (Cluster I) and genetic type-II (Cluster 2) from previous studies [21,31]. Within Cluster 1, it could further be divided into two groups, which we tentatively referred to as genetic type-Ia (Cluster Ia) and genetic type-Ib (Cluster Ib). The Malaysian isolates of GP/UPM/MY001, GP/UPM/MY002, GP/UPM/MY003 and GP/UPM/MY004 were closely related to the USA strain 685-99, the Germany strain 84-15995, China strains Hs-DY05 and F603, and clustered among genetic type Ia group. On the other hand, isolates GP/UPM/MY007, GP/UPM/MY005, GP/UPM/MY008, GP/UPM/MY009, GP/UPM/MY010 and GP/UPM/MY011 were more closely related to China strain SW124 and USA strain MN-H; clustered among genetic type Ib. There was only one isolate that was clustered among genetic type-II, that was isolate GP/UPM/MY006, which was most closely related to China strain LHDR_HPS_1_2.

The further comparison of subclusters (Cluster Ia and Cluster Ib) and clusters (Cluster I and II) revealed a homology of 95.2–98.9% and 92.98–97.32%. All sequences in Cluster II generally have a longer nucleotide length (1167–1203) as compared to Cluster I (1077–1095). Information on each isolate, and their results, were summarized in Table 1.

## 4. Discussion

Although serotyping provides only part of the information on the virulence of *G. parasuis*, it is important in terms of selecting the best vaccine for control. This is because current vaccine employs the inactivation/attenuation of bacteria whole cell from one or two serotypes, which might not produce desirable protection to heterologous challenge. Currently available commercial vaccines targets serotypes 4, 5 and 12; 1 and 6, or serotype 5 only [35], while the only approved commercial vaccine in Malaysia targets serotype 3, 4 and 5 [36]. When compared to the results of this study, vaccines which may be suitable would most likely be those against the highly virulent serotype 5 or 12 strains, and to a lesser extent those against serotype 8 and 15. Nonetheless, the majority of the isolates described in this study, which were obtained from two major pig producing states in Malaysia (Perak and Selangor), were detected to be serotype 5 or 12. This is consistent to the predominance of serotype 5 or 12 found in North America, Europe and Asia [37]. For a better overview of the *G. parasuis* serotypic distribution within the country, the current study may be expanded to include all pig producing states in Malaysia.

The upregulation of *vtaA* proteins during infection supports the role of the protein in infection, especially by virulent strains [14]. This is coherent to findings where virulent *vtaA* was detected in all strains isolated from clinical sites (lung, pleural wall joint, brain and peritoneal wall), and nonvirulent *vtaA* was detected in one strain that was isolated from the carrier site (tonsil). This demonstrated that the difference in *vtaA* might be due to isolation sites, whether *G. parasuis* was isolated from carrier or systemic sites, which was similar to findings in this study [12].

After the Nipah virus outbreak in Malaysia (1998–1999), there is a lack of national breeder units within the country [38]. Since then, Malaysia has been importing live pigs, for breeding purposes, from Canada, Denmark, Finland, France, South Korea, Sweden, the United Kingdom and the USA [39]. However, the team could not find an agreement to the point where the findings of closely related strains were due to direct transmission from imported breeder animals to the farm, as Malaysian isolates were found to be closely related to Germany and China. This is highly possible due to the lack of the *OmpP2* gene sequence information deposited by the country that Malaysia is importing from, as well as from neighboring countries. Nevertheless, as only one gene was used for comparison, the information gained was limited.

In this study, strains in Cluster I (genetic type-I) had a shorter *OmpP2* gene nucleotide length, which mainly displayed virulent *vtaA,* and the opposite was seen in strains of Cluster II. The shorter length of *OmpP2* proteins was demonstrated to exhibit significantly increased resistance to complement killing, complementing the idea that genetic type I strains are likely to be causing disease [40]. The detection of virulent *vtaA* in Malaysia isolates belonging to *OmpP2* genetic type I, and nonvirulent *vtaA* in isolates belonging to *OmpP2* genetic type II, seems to be parallel. However, mapping information from previous literature [41] showed that strains from genetic type II, such as strain H465 and SW124 of Germany, also possess virulent *vtaA*, even though all strains in the genetic type I were found to possess virulent *vtaA*. This may indicate that the two genes are not directly related, but when combined, both may yield more severe clinical infections; this is worth being studied further.

## 5. Conclusions

In conclusion, this study reveals that more than 91% of the isolates isolated from pigs suffering from Glässer’s disease belong to the *OmpP2* genotype 1 cluster, and that a substantial proportion of disease (91%) was caused by *vtaA* virulent strains.

## Figures and Tables

**Figure 1 pathogens-12-00308-f001:**
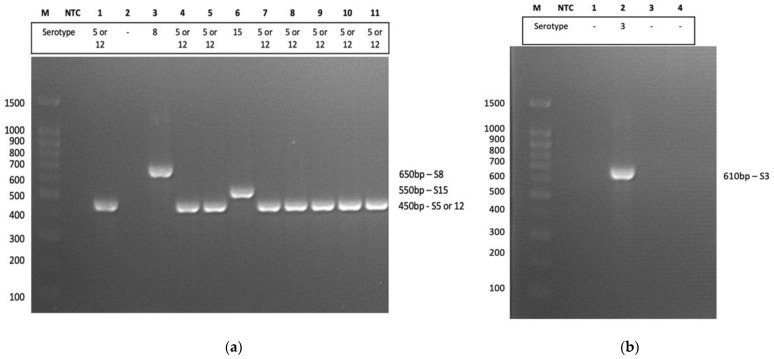
Serotyping using primers by Howell et al. (2015) [4]: (**a**) with an adopted method of the set 2 recommendation by Schuwerk et al., (2020) [12]. M, molecular weight marker (100 bp); NTC, nontemplate control; 1, GP/UPM/MY001; 2, GP/UPM/MY002; 3, GP/UPM/MY003; 4, GP/UPM/MY004; 5, GP/UPM/MY005; 6, GP/UPM/MY006; 7, GP/UPM/MY007; 8, GP/UPM/MY008; 9, GP/UPM/MY008; 10, GP/UPM/MY010; 11, GP/UPM/MY011; (**b**) with the set 1 recommendation by Schuwerk et al., (2020) [12]. M, molecular weight marker (100 bp); NTC, nontemplate control; 1, GP/UPM/MY001; 2, GP/UPM/MY002; 3, GP/UPM/MY003; 4, GP/UPM/MY004.

**Figure 2 pathogens-12-00308-f002:**
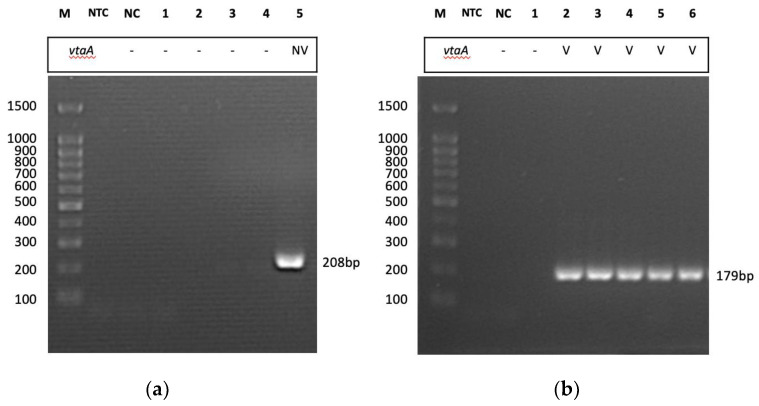
A PCR gel photo of *vtaA* amplification. (**a**) M, molecular weight marker (100 bp); NTC, nontemplate control; NC, negative control (*S. suis*); 1–4, samples yielded poor result that were not including in this publication; 5, GP/UP/MY006; (**b**) M, molecular weight marker (100 bp); NTC, nontemplate control; NC, negative control (*S. suis*); 1, sample yielded negative result that was not including in this publication; 2, GP/UPM/MY001; 3, GP/UPM/MY002; 4, GP/UPM/MY003; 5, GP/UPM/MY004; 6, GP/UPM/MY005; V; virulent.

**Figure 3 pathogens-12-00308-f003:**
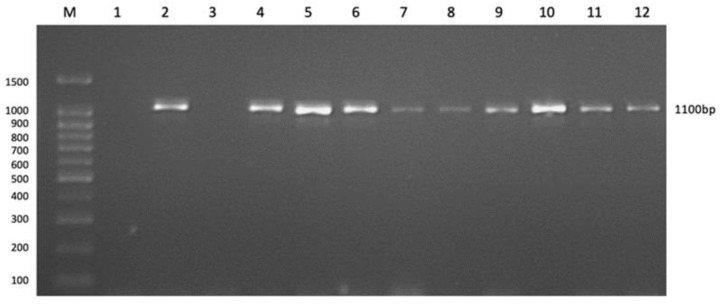
An electrophoresed gel photo for the amplification of *OmpP2* gene (GP/UPM/MY011 not shown here). M, molecular weight marker (100 bp); 1, nontemplate control; 2, GP/UPM/MY001; 3, negative control (*Streptococcus suis*); 4, GP/UPM/MY002; 5, GP/UPM/MY003; 6, GP/UPM/MY004; 7, GP/UPM/MY005; 8, GP/UPM/MY006; 9, GP/UPM/MY007; 10, GP/UPM/MY008; 11, GP/UPM/MY008; 12, GP/UPM/MY010.

**Figure 4 pathogens-12-00308-f004:**
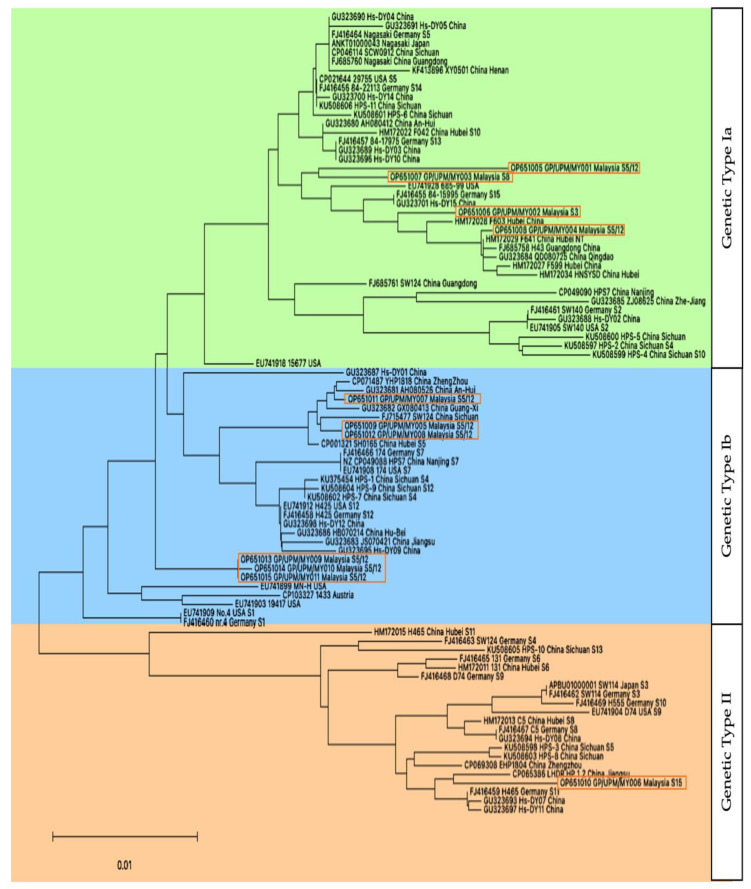
A phylogenetic tree of the *Glaesserella parasuis OmpP2* gene nucleotide sequences. Evolutionary history was inferred using the neighbor-joining method [32]. The optimal tree is shown. The tree is drawn to scale, with branch lengths in the same units as those of the evolutionary distances used to infer the phylogenetic tree. The evolutionary distances were computed using the maximum composite likelihood method [33] and are in the units of the number of base substitutions per site. This analysis involved 90 nucleotide sequences, i.e., 11 Malaysian *G. parasuis OmpP2* gene nucleotide sequences and 79 reference sequences from the NCBI GenBank. All ambiguous positions were removed for each sequence pair (pairwise deletion option). There were a total of 1299 positions in the final dataset. Evolutionary analyses were conducted in the MEGA 11 software [29,34]. The GenBank accession numbers, strain, origin country/location and serotype (if available) are as indicated. Malaysian gene sequences were additionally highlighted with a box.

**Table 1 pathogens-12-00308-t001:** A summary of isolate information and results.

Isolate ID	State	Farm	Animal	Year	Farm Size (Sow Number)	Postmortem Lesion	Site of Isolation	GenBank Accession Number	Serotype	*vtaA*	Genetic Type
GP/UPM/MY001	Selangor	S1	S1001	2018	1000	Polyserositis, arthritis	lung	OP651005	5 or 12	Virulent	Ia
GP/UPM/MY002	Selangor	S2	S2001	2018	300	Arthritis	joint	OP651006	3	Virulent	Ia
GP/UPM/MY003	Selangor	S3	S3001	2018	300	Fibrinous pneumoniae	lung	OP651007	8	Virulent	Ia
GP/UPM/MY004	Selangor	S3	S3002	2018	300	Fibrinous pneumoniae	lung	OP651008	5 or 12	Virulent	Ia
GP/UPM/MY005	Selangor	S4	S4001	2020	900	Polyserositis	brain	OP651009	5 or 12	Virulent	Ib
GP/UPM/MY006	Selangor	S4	S4001	2020	900	Polyserositis	tonsil	OP651010	15	Nonvirulent	II
GP/UPM/MY007	Perak	P1	P1001	2020	500	Polyserositis	brain	OP651011	5 or 12	Virulent	Ib
GP/UPM/MY008	Selangor	S5	S5001	2021	400	Polyserositis	pleural swab	OP651012	5 or 12	Virulent	Ib
GP/UPM/MY009	Perak	P2	P2001	2022	4000	Polyserositis, arthritis	lung	OP651013	5 or 12	Virulent	Ib
GP/UPM/MY010	Perak	P2	P2001	2022	4000	Polyserositis, arthritis	peritoneal swab	OP651014	5 or 12	Virulent	Ib
GP/UPM/MY011	Perak	P2	P2001	2022	4000	Polyserositis, arthritis	joint	OP651015	5 or 12	Virulent	Ib

## Data Availability

The Malaysian *Glaesserella parasuis OmpP2* sequences reported in this study have been deposited at GenBank (http://www.ncbi.nlm.nih.gov) under accession numbers OP651005–OP651015.

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
