# Peer review of "Molecular Characterization and Phylogenetic Analysis of *Outer membrane protein P2* (*OmpP2*) of *Glaesserella* (*Haemophilus*) *parasuis* Isolates in Central State of Peninsular Malaysia"

_pathogens, 2023, doi:10.3390/pathogens12020308_

Round 1

Reviewer 1 Report

Dear editor, 

The authors of this manuscript used conventional methods described earlier by other researchers to analyse 11 isolates of G. parasuis taken from different states, organs and times in Malaysia. 

The manuscript requires more work and improvement especially in the materials and methods section and the results. 

The Number of isolates taken is not sufficient to make the conclusions the authors made. More information is required about the overall situation in the country, and how many isolates they had in the beginning to show if those 11 isolates are representative of the situation or not. 

I make my comments clear in the manuscript in the uploaded pdf file. 

Please find it attached in this letter.

Author Response

Attached also is a copy of the edited manuscript with track changes for your reference.

Response to Reviewer 1 Comments

Point 1: The manuscript requires more work and improvement especially in the materials and methods section and the results.

 Response 1: Thank you for the suggestion, we have improved on the materials and methods as well results section by adding more detail to clarify on the sampling/isolates (Farm size of the samples collected and animal involved) and method used.

Point 2: The number of isolates taken is not sufficient to make the conclusions the authors made. More information is required about the overall situation in the country, and how many isolates they had in the beginning to show if those 11 isolates are representative of the situation or not.

Response 2: Thank you for the comment.

We added the following statement to highlight overall situation in the country and our samples collection process

Line 101-106

The central states of Perak and Selangor contributes the highest pig production in Peninsular Malaysia. Farm size from both states ranges from 100 to 4000 sow number.  Due to the pandemic and threat of African Swine Fever, access to farms or farms submitting samples specifically for bacteria isolation was limited. There were twenty-eight (28) Glaesserella parasuis isolates from central states of Peninsular Malaysia were randomly selected retrievable from archive samples (2018-2020; n=24) and recent clinical cases (2021-2022; n=4) presented to the Swine Unit of Faculty of Veterinary Medicine, Universiti Putra Malaysia. The isolation of G. parasuis from clinical samples were only performed upon special request by farmers and not a common routine procedure. Isolation of G. parasuis was done by inoculating samples submitted on plated chocolate agar (Oxoid, USA), identified by conventional PCR and positive isolates were preserved at 10% skim milk kept in -80°C.

In addition, we also added “Central State of” to the title to reflect the actual sampling area, the new title is as below ‘Molecular characterization and phylogenetic analysis of Outer membrane protein P2 (OmpP2) of Glaesserella (Haemophilus) parasuis Isolates in Central State of Peninsular Malaysia’.

Point 3: I make my comments clear in the manuscript in the uploaded pdf file. Please find it attached in this letter.

 Response 3: Thank you for the comments, we made the amendment as follows:

Reviewer 1 comments in file

Amendments made

  1. Abstract line 26-27

Rephrasing

Removed “possibly related to the site of isolation” as authors feel not appropriate to make such conclusion after further considerations.

  1. 2.3 Conventional PCR assay for vtaA gene identification

How are the results analysed

Detail method was described and further explain at Results supported with figure.

  1. 3. Results
    Please write a small section in the results about culturing and how the isolates were chosen. You mentioned it was Random, please specify, the total number of isolates per state, and how those 11 isolates were chosen.

Statement on overall situation in the country and our samples collection process was added.

  1. Line 147: Please rephrase this sentence

    …primers (Table 1) and the results are shown in Figure 1a and 1b

Rephrased. Table 1 merged to the existing table 2 as commented by Reviewer 2.

  1. Line 158: Delete the word “detected”

Deleted.

  1. Line 160-164
    Did you test several samples from one animal? This is what I understood from this paragraph, while in the table 1, you mentioned that each isolate was taken from one organ.

    If this is the case, then please show the results how you studied several samples from one animal, and how serotypes were isolates from specific organs, and please rewrite the Materials and methods, mentioning that you took from this pig several samples.

Further elaborated at the result part as suggested at comment 3 and detail information listed at the new Table 1.

  1. Line 168-169
    Did you test tonsils that showed negative results? what is your control in this case?

No. Available sample(tonsil) were limited to those collected from clinical cases/ or clinically ill animals.

  1. Line 175-176
    This part is well written in the next section using percent of homology, please remove "Highly specific"

Word removed.

  1. Line 185
    Which sequencer was used in this part, and what is the kit used? A fragment of 1100 bp might not always be easily sequenced by Sanger, and might require two overlapping pairs of primers. Please explain this more in the Materials and Methods and in this part.

This is further explained in the Materials and Method, 2.4. PCR Amplification and bioinformatic analysis of OmpP2 gene.

In short it was outsourced to sequence for the forward and reverse strand, consensus sequences were acquired for the analysis.

Line 222
This depends on the original number of isolates in the. country. The studied isolates (n=11) might not be representative of the overall situation. I recommend not to make such deductions.

Amended to:

Nonetheless, majority of the isolates described in this study, which were obtained from 2 major pig producing states in Malaysia (Perak and Selangor), were detected to be serotype 5/12. This is consistent to the predominance of serotype 5/12 found in North America, Europe, and Asia [34]. The current study may be expanded to include all pig producing states in Malaysia for a better overview of the G. parasuis serotypic distribution within the country.

Figure 3. Please provide sharper image

New sharper image attached. Please note a slight layout change with colour change to enhance clarity or image, no data was altered.

Reviewer 2 Report

Yien-Lee and cols., evaluated the molecular characteristics of Glaesserella parasuis isolated in Malaysia. The authors evaluated 11 isolated of G. parasuis from clinically sick pigs collected from the main pig-producing regions in Malaysia. By using PCR, the authors classified the isolated as serotypes 3, 5, 8, 12, and 15. According to the authors, serotypes 3, 8, and 15 were never reported in Malaysia. Besides, the authors analyzed the phylogenetic characteristics of the OmpP2 gene and showed that the majority belong to genotype 1 and one isolate to genotype 2. 

In general, the results of this study are limited. Only 11 isolates were evaluated and, in my opinion, are not representative of Malaysia. In my opinion, the results of this manuscript do not have the impact for a high-impact journal such as Pathogens. If the authors can information such as the number of samples positives since 2018 or add the number of isolations, the manuscript's impact will increase. As the authors mentioned, the results are limited to support the classification as "virulent". 

The introduction is long; I recommend reducing it and focusing on the main aspects of serovars of G. parasuis and previous studies in Asia and other regions.

The results of section 3.1.2 are missing

Figure 3 is low quiality

Author Response

Response to Reviewer 2 Comments

Point 1: In general, the results of this study are limited. Only 11 isolates were evaluated and, in my opinion, are not representative of Malaysia. In my opinion, the results of this manuscript do not have the impact for a high-impact journal such as Pathogens. If the authors can information such as the number of samples positives since 2018 or add the number of isolations, the manuscript's impact will increase. As the authors mentioned, the results are limited to support the classification as "virulent".  

Response 1: Thank you for the comment. We added the following statement to highlight overall situation in the country and our samples collection process

Line 101-106

The central states of Perak and Selangor contributes the highest pig production in Peninsular Malaysia. Farm size from both states ranges from 100 to 4000 sow number.  Due to the pandemic and threat of African Swine Fever, access to farms or farms submitting samples specifically for bacteria isolation was limited. There were twenty-eight (28) Glaesserella parasuis isolates from central states of Peninsular Malaysia were randomly selected retrievable from archive samples (2018-2020; n=24) and recent clinical cases (2021-2022; n=4) presented to the Swine Unit of Faculty of Veterinary Medicine, Universiti Putra Malaysia. The isolation of G. parasuis from clinical samples were only performed upon special request by farmers and not a common routine procedure. Isolation of G. parasuis was done by inoculating samples submitted on plated chocolate agar (Oxoid, USA), identified by conventional PCR and positive isolates were preserved at 10% skim milk kept in -80°C.

In addition, we also added “Central State of” to the title to reflect the actual sampling area, the new title is as below ‘Molecular characterization and phylogenetic analysis of Outer membrane protein P2 (OmpP2) of Glaesserella (Haemophilus) parasuis Isolates in Central State of Peninsular Malaysia’.

Point 2: The introduction is long; I recommend reducing it and focusing on the main aspects of serovars of G. parasuis and previous studies in Asia and other regions.

Response 2: Thank you for your comment.

Content in line 69-72 and 86-90 were removed to make the introduction more condense.

We also modified sentences at line 53 to make the statement more precise.

Point 3: The results of section 3.1.2 are missing

Response 3: Thank you for your comment.

The results are presented as figures 2(a) and 2(b). Additionally, we rename the section 3.1.2 to section 3.3 and renumbered the rest of the result section to correspond to the material and method section for easier reference.

Point 4: Figure 3 is low quality

Response 4: Thank you for your comment.

New sharper image attached. Please note a slight layout change with colour change to enhance clarity or image, none data was altered.

Reviewer 3 Report

In this manuscript authors tried to characterize the 11 strains of G. parasuis isolate from diseased pigs from two central states of the country, regarding: 1.  serotype. 2. pathotype, based on LS-PCR on vtaA gene. 3. genotype, based on OmpP2 gene.

1. the scale of study and the number of bacterial isolates is low, this manuscript should be more appropriate a "brief communication".

2. information of Tables 1 and 2 should be merged into a single large Table for easy reading at a glance.  There is no need to separate into two. It is also clear that LS-PCR, although very good, is not absolute, because you can find the so-called  non-virulent isolate (006) actually causing polyserositis. The genotype is also not absolute for judging pathogenicity.

3. line 218: the serotype and genotype of vaccine should be provided, since you discuss about it and compared with the disease strains.

Author Response

Response to Reviewer 3 Comments

Point 1: the scale of study and the number of bacterial isolates is low; this manuscript should be more appropriate a "brief communication".

Response 1: Thank you for your comment.

Line 101-106

The central states of Perak and Selangor contributes the highest pig production in Peninsular Malaysia. Farm size from both states ranges from 100 to 4000 sow number. Due to the pandemic and threat of African Swine Fever, access to farms or farms submitting samples specifically for bacteria isolation was limited. There were twenty-eight (28) Glaesserella parasuis isolates from central states of Peninsular Malaysia were randomly selected retrievable from archive samples (2018-2020; n=24) and recent clinical cases (2021-2022; n=4) presented to the Swine Unit of Faculty of Veterinary Medicine, Universiti Putra Malaysia. The isolation of G. parasuis from clinical samples were only performed upon special request by farmers and not a common routine procedure. Isolation of G. parasuis was done by inoculating samples submitted on plated chocolate agar (Oxoid, USA), identified by conventional PCR and positive isolates were preserved at 10% skim milk kept in -80°C.

In addition, we also added “Central State of” to the title to reflect the actual sampling area, the new title is as below ‘Molecular characterization and phylogenetic analysis of Outer membrane protein P2 (OmpP2) of Glaesserella (Haemophilus) parasuis Isolates in Central State of Peninsular Malaysia’.

Point 2: information of Tables 1 and 2 should be merged into a single large Table for easy reading at a glance.  There is no need to separate into two. It is also clear that LS-PCR, although very good, is not absolute, because you can find the so-called non-virulent isolate (006) actually causing polyserositis. The genotype is also not absolute for judging pathogenicity.

Response 2: Thank you for your comment.

Thanks for the suggestion, we already merge the Table 1 and Table 2 together and rename as Table 1.

Point 3: the serotype and genotype of vaccine should be provided, since you discuss about it and compared with the disease strains.

Response 3: Thank you for your comment.

The serotypes of the currently available vaccine was elaborated.

Round 2

Reviewer 1 Report

Dear Editor, 

I would like to thank the authors for answering my comments.

Reviewer 2 Report

Thank you the authors, for the response to my comments. 

I still think that results are limited, maybe at the borderline, to decide reject or not. 

I have no additional observations.